# Cereals and Fruits as Effective Delivery Vehicles of *Lacticaseibacillus rhamnosus* through Gastrointestinal Transit

Grigorios Nelios [1,*], Ioanna Prapa [1], Anastasios Nikolaou [1], Gregoria Mitropoulou [1], Amalia E. Yanni [2], Nikolaos Kostomitsopoulos [3] and Yiannis Kourkoutas [1,*]

1   Laboratory of Applied Microbiology and Biotechnology, Department of Molecular Biology and Genetics, Democritus University of Thrace, Dragana, 68100 Alexandroupolis, Greece
2   Laboratory of Chemistry, Biochemistry, Physical Chemistry of Foods, Department of Nutrition and Dietetics, Harokopio University of Athens, 17671 Athens, Greece
3   Laboratory Animal Facility, Biomedical Research Foundation of the Academy of Athens, 11527 Athens, Greece
*   Correspondence: gnelios@mbg.duth.gr (G.N.); ikourkou@mbg.duth.gr (Y.K.); Tel.: +30-2551-030-703 (G.N.); +30-25-5103-0633 (Y.K.)

**Abstract:** The viability of probiotic cells during their transit through the degradative conditions of the gastrointestinal tract is considered an essential prerequisite for their effectiveness. To enhance the survival of probiotics, cell immobilization has been proposed as a promising strategy, creating a protective microenvironment. In the present study, the viability of immobilized *Lacticaseibacillus rhamnosus* OLXAL-1 cells on cereals and fruits was investigated in comparison to free cells, applying both an in vitro static digestion and an in vivo mouse model. During the in vitro digestion, the survival rates of all immobilized *L. rhamnosus* OLXAL-1 cultures were higher compared to free cells, with the highest survival rate recorded in oat flakes (84.76%). In a subsequent step, following the administration of both immobilized and free cells to BALB/c mice, a significant increase in lactobacilli populations was observed in the mice feces compared to baseline. Notably, the group receiving the immobilized cells exhibited significantly higher lactobacilli counts compared to the group fed with free cells (8.02 log CFU/g and 7.64 log CFU/g, respectively). Finally, the presence of *L. rhamnosus* cells at levels > 6 log CFU/g was verified in the mice feces in both groups through multiplex PCR analysis.

**Keywords:** probiotics; cell immobilization; in vitro digestion; BALB/c mouse model; gastrointestinal transit; cereals; fruits

## 1. Introduction

Over the past two decades, the study of the human microbiome has rapidly emerged as an attractive area of research, aiming at providing novel insights into complex ecosystems [1]. The microbiome, according to the most cited definition [2], is described as "a community of commensal, symbiotic and pathogenic microorganisms within a body space or other environment". However, contemporary definitions point out that the term microbiome does not only refer to the microorganisms involved, but also encompasses their "theatre of activity" [3], including the formation of a dynamic and interactive micro-ecosystem prone to change in time and scale. Additionally, it involves integration into macro-ecosystems, including eukaryotic hosts [1]. Probiotics have been identified as potentially effective therapeutic agents, with the capacity to improve health by modifying the microbiome composition and functionality of the gut ecosystem. Probiotics, defined by the World Health Organization and Food and Agriculture Organization (WHO/FAO) as "live microorganisms which when administered in adequate amounts confer a health benefit on the host" [4], have prompted substantial research efforts to identify their mechanisms of action and maximize their efficacy.

One of the fundamental aspects influencing the effectiveness of probiotics is the viability and activity of the administered cells [5]. The survival of probiotic cells during

their transit through the gastrointestinal (GI) tract is pivotal for exerting their intended health effects. Factors like exposure to harsh gastric conditions, bile salts, and enzymatic activities can significantly impact the viability and functionality of probiotic cells [6,7]. Thus, ensuring their viability throughout GI transit becomes a critical consideration in developing effective probiotic formulations.

Cell immobilization has emerged as a promising strategy to enhance the viability and activity of probiotic cells. This technique involves the physical or chemical adhesion of live cells onto a support matrix, providing a protective microenvironment that shields the cells from harsh conditions [8]. Various carriers have been explored for immobilizing probiotic cells, including natural polymers [9], gels [10], fibers [11] and food matrices [12]. These immobilization approaches have been reported to offer several advantages, such as increased cell stability, prolonged shelf-life, improved resistance to environmental stresses, faster fermentation rates and reduced risk for microbial contamination [13].

Despite the growing body of research focusing on finding novel probiotic strains, few studies have rigorously compared in vitro testing of simulated GI transit with more realistic in vivo models. Instead, the majority of studies evaluate the tolerance of probiotic strains by applying separated static in vitro assays, which do not adequately replicate the sequential stress conditions encountered in vivo [14]. Indeed, the use of in vivo models, such as murine models, is still crucial for comprehensively evaluating the efficacy of probiotic strains, due to the presence of dynamic interactions between probiotics and the host's GI system [15].

In the present study, the presumptive probiotic *Lacticaseibacillus rhamnosus* OLXAL-1 strain, isolated from Greek olive fruits (*Olea europea* var. *rotunda*) and previously described as a candidate agent to manage type 1 diabetes [16], was selected. The viability of immobilized *L. rhamnosus* OLXAL-1 cells on natural food ingredients was investigated in comparison to free cells using an in vitro static digestion model. Additionally, GI transit was further validated in an in vivo mouse model, thereby establishing a more realistic framework for evaluating probiotic efficacy.

## 2. Materials and Methods

### 2.1. Bacterial Strain and Culture Conditions

*Lacticaseibacillus rhamnosus* OLXAL-1, isolated from Greek olive fruits [16], was grown on a sterilized food-grade medium, as previously described by Nikolaou et al. [17].

### 2.2. Cell Immobilization on Natural Supports and Production of Freeze-Dried Cultures

Initially, grown *Lacticaseibacillus rhamnosus* OLXAL-1 cells were obtained by centrifugation ($8500 \times g$ for 10 min at 4 °C), subsequently rinsed with sterile ¼-strength Ringer's solution (VWR International GmbH, Radnor, PA, USA), and finally harvested by centrifugation (wet free cells).

For cell immobilization, the collected cells were resuspended in a sterile ¼-strength Ringer's solution to achieve the initial culture volume (immobilization solution) [17]. As immobilization carriers, four natural supports were selected, including oat and wheat flakes and apple (Granny Smith cultivar) and banana (Giant Cavendish cultivar) pieces. Before the immobilization process, oat and wheat flakes were heated in an oven at 140 °C for 30 min, while apple and banana pieces were used with no pretreatment. Oat and wheat flakes were immersed in the immobilization solution (in a ratio of 30% *w/v*) for 15 min, while apple ($0.4 \pm 0.1$ cm side length) and banana pieces ($1.0 \pm 0.2$ cm side length) in a ratio of 60% *w/v* for 4 h and 6 h, respectively. During cell immobilization, the natural supports were left undisturbed at room temperature. After the completion of cell immobilization, the natural supports were strained and subjected to rinsing with sterile ¼-strength Ringer's solution, aiming to remove any non-immobilized cells (wet immobilized cells).

Freeze-dried immobilized cells on natural supports were produced as previously described [18]. For comparison purposes, wet free *L. rhamnosus* OLXAL-1 cells were also subjected to a 24 h freeze-drying process (freeze-dried free cells).

### 2.3. Scanning Electron Microscopy

The immobilization of *L. rhamnosus* OLXAL-1 cells on cereals and fruit pieces was assessed through scanning electron microscopy following the methodology described by Kourkoutas et al. [19]. In brief, freeze-dried immobilized cultures were coated with gold using a BAL-TEC MED 020 Sputter coating system for 2 min, and subsequently examined using a JSM-6300 scanning electron microscope at 20 kV voltage.

### 2.4. Static In Vitro Digestion Model

For the in vitro evaluation of the survivability of freeze-dried immobilized *L. rhamnosus* OLXAL-1 culture on natural carriers, a static digestion simulation model was developed based on the studies of Minekus et al. [20] and Madureira et al. [21] (Figure 1). Specifically, 20 g of rehydrated immobilized cultures on natural carriers was homogenized using an iMix bag mixer (Interlab, Mourjou, France), added to 20 mL of simulated salivary fluid (SSF) [20] containing 75 U/mL α-amylase from *Bacillus* spp. (Type-IIA, ≥1500 U/mg) (Sigma-Aldrich, St Louis, MO, USA) and 50 U/mL lysozyme from chicken egg white (≥20,000 U/mg) (Apollo Scientific, Cheshire, UK), and incubated at 37 °C for 2 min (simulated salivary phase). For the rehydration of immobilized cultures, 20 g of freeze-dried immobilized *L. rhamnosus* OLXAL-1 cells on natural carriers was submerged in 80 mL of sterilized dH$_2$O at room temperature for 30 min and subsequently strained.

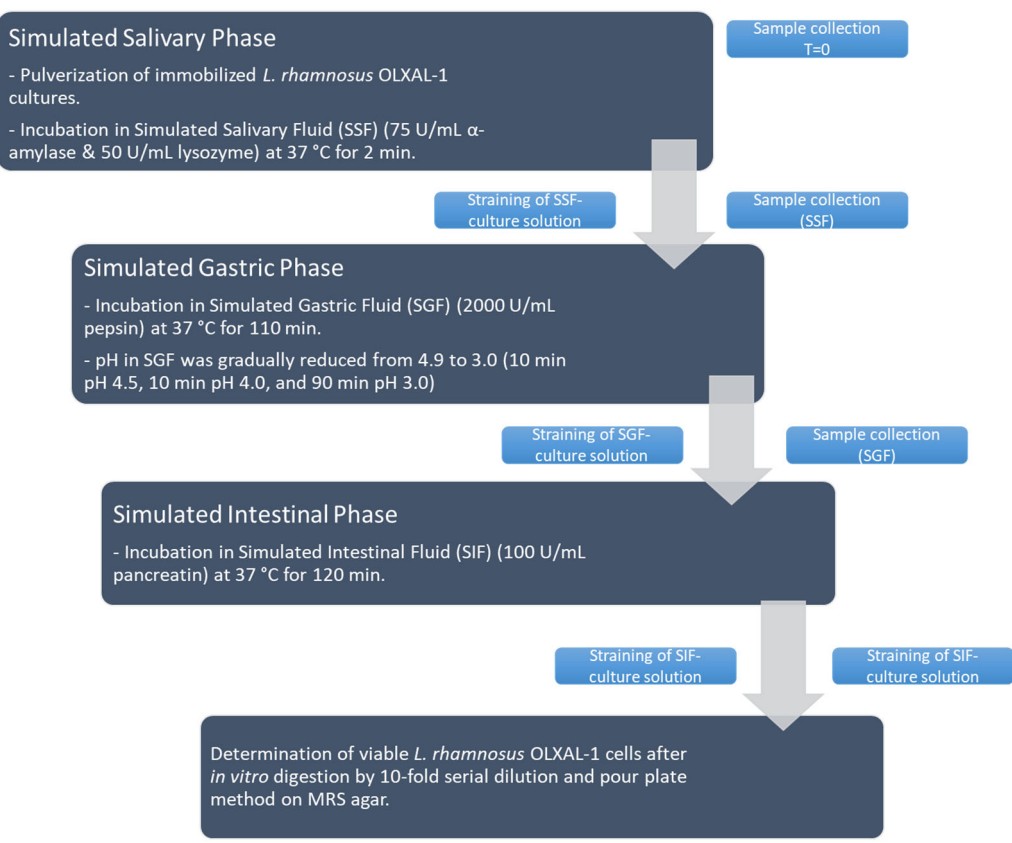

**Figure 1.** Schematic representation of the static digestion simulation model.

After the simulated salivary phase, the culture-SSF solution was strained, the liquid phase was discarded and then 10 g of immobilized culture was added to 10 mL of simulated gastric fluid (SGF) [20], containing 2000 U/mL pepsin from porcine gastric mucosa (≥3200 U/mg) (Sigma-Aldrich, St Louis, MO, USA), and incubated at 37 °C for 110 min (simulated gastric phase). During incubation, the pH of the immobilized culture-SGF solution gradually decreased from 4.9 to 3.0, aiming to better simulate the conditions of food transition from the oral cavity to the stomach [20].

Following the simulated gastric phase, the culture solution-SGF was strained, the liquid phase was discarded, and then 5 g of immobilized culture was added to 5 mL of simulated intestinal fluid (SIF) solution [20], containing 100 U/mL porcine pancreatic mucin (8 × USP specifications) (Sigma-Aldrich, St Louis, MO, USA), and incubated at 37 °C for 120 min (simulated intestinal phase).

After the simulated intestinal phase, the immobilized culture-SSF solution was strained, the liquid phase was discarded, and the immobilized culture was collected.

Throughout all three simulated digestion phases, enzyme solutions were prepared fresh daily, and the pH was adjusted by adding either 0.5 M HCl or 1 M $NaHCO_3$. Samples of immobilized *L. rhamnosus* OLXAL-1 cultures were collected immediately following the addition of SSF and at the completion of each simulated digestion phase. Viable *L. rhamnosus* cell counts were assessed using a 10-fold serial dilution and pour plate method on MRS agar (Condalab, Madrid, Spain). The MRS plates were anaerobically incubated at 37 °C for a minimum of 72 h. The survival rates were subsequently determined in accordance with Nelios et al. [16].

### 2.5. Animals and In Vivo Study Design

A total of twenty four 8-week-old male BALB/c mice (weighing 20–25 g bw), bred in the Laboratory Animal Facility of the Biomedical Research Foundation of the Academy of Athens (BRFAA), were used. The housing conditions for the mice were in accordance with the specifications previously outlined by Kompoura et al. [22].

Mice were allocated into four groups in a random manner, taking into account the different dietary treatments: (a) animals that followed the control diet (n = 6, CD) or (b) the control diet supplemented with a daily dose of 2 g of freeze-dried oat flakes (n = 6, OD), and (c) animals that received the control diet supplemented with a daily dose of 2 g of freeze-dried immobilized *L. rhamnosus* OLXAL-1 cells on oat flakes (n = 6, ID), or (d) the control diet supplemented with freeze-dried free *L. rhamnosus* OLXAL-1 cells (n = 6, FD). In both the ID and FD groups, all animals received the same daily dose of $2 \times 10^9$ CFU, with the aim of evaluating the effect of immobilization on cell viability compared to free cells. The dietary intervention lasted for 10 days and was based on the previous study by Kuda et al. [23]. The control diet included a daily portion of 5 g of mouse chow, while the control diet supplemented with freeze-dried oat flakes included a daily portion of 3 g of mouse chow plus 2 g of freeze-dried oat flakes. Each morning, the animals were supplied with fresh food according to the above-mentioned diets, while any unconsumed chow was promptly cleared from the cages. In all cases, fecal samples from each animal were collected before and after the 10-day dietary intervention. Upon completion of the experimental period, animals were euthanized in a random manner by administering an overdose of isoflurane (ISOVET, Chanelle Pharma, Loughrea, Co., Galway, Ireland). Following euthanasia, intestinal fluid and tissue samples from the small and large intestine (ileum, cecum and colon) were also collected and subjected to microbiological and molecular analysis. Upon collection, intestinal tissue and intestinal fluid samples were diluted in 25% glycerol-Ringer's solution (1:1 ratio) and then preserved at a temperature of −80 °C until further microbiological analysis [22]. The animal experimentation was reviewed by the Project Evaluation Committee of the BRFAA and approved by the Veterinary Directorate of the Athens Prefecture (Ref. Number 483166/30-05-2022) in compliance with National Legislation and European Directive 2010/63.

### 2.6. Microbiological Analysis and Enumeration of Lactobacilli

Viable lactobacilli cell counts in the fecal and intestinal fluid samples were assessed by 10-fold serial dilution and by both pour and spread plate methods on MRS agar (Condalab, Madrid, Spain). MRS plates were incubated anaerobically at 37 °C for at least 72 h. Lactobacilli counts were determined accordingly in intestine tissue samples in which an additional step was applied where samples underwent thorough vortex mixing to disrupt bacterial clumps and detach loosely bound bacteria [24].

### 2.7. Identification of L. rhamnosus OLXAL-1 by Multiplex PCR

In order to verify the presence of *L. rhamnosus* cells in the fecal and intestinal samples after the growth of the lactobacilli cultures on MRS agar, the spread method plates were washed with 3 mL of sterile ¼-strength Ringer's solution; subsequently, the cell suspension was subjected to molecular analysis based on multiplex PCR methodology [25]. The extraction of genomic DNA from the lactobacilli suspensions was conducted utilizing a Nucleospin tissue kit (Macherey-Nagel, Düren, Germany), according to the manufacturer's instructions. Amplification reactions were carried out on a DNA Engine, Peltier Thermal Cycler (Bio-Rad Laboratories, Hercules, CA, USA). Analysis of the PCR products was performed by electrophoresis using a 2% agarose gel in 1% tris-borate-EDTA (TBE) buffer, and visualization was achieved through UV transillumination (Bio-Rad Laboratories, Hercules, CA, USA). *Lacticaseibacillus rhamnosus* GG (ATCC 53103) was utilized as a reference strain.

### 2.8. Statistical Analysis

Values are presented as the mean ± SD. Statistical analysis was conducted using Statistica v. 12 statistical software (StatSoft, Inc., Tulsa, OK, USA). Differences in mean values were evaluated using one-way and two-way ANOVA, followed by the Bonferroni post-hoc test, with a significance level set at 0.05.

## 3. Results and Discussion

### 3.1. Immobilized Lacticaseibacillus rhamnosus OLXAL-1 Cultures

At first, free and immobilized *L. rhamnosus* OLXAL-1 cells on cereals (oat and wheat flakes) and fruits (apple and banana pieces) were produced in both wet and freeze-dried form, and, subsequently, viable *L. rhamnosus* OLXAL-1 cell counts were determined (Figure 2). The levels of wet immobilized cultures on oat and wheat flakes ranged from 9.08 to 9.19 log CFU/g, while the corresponding levels of apple and banana pieces were > 8 log CFU/g. After freeze-drying, the levels of freeze-dried immobilized cells on oat and wheat flakes were >9 log CFU/g (dry cell mass) and >8 log CFU/g (wet cell mass), while the corresponding levels in the apple and banana pieces were >8 log CFU/g (dry cell mass) and >7 log CFU/g (wet cell mass). Notably, the daily minimum recommended cell concentration [26,27], in order to deliver beneficial health effects on the host, was achieved in all immobilized freeze-dried cell cultures on natural carriers.

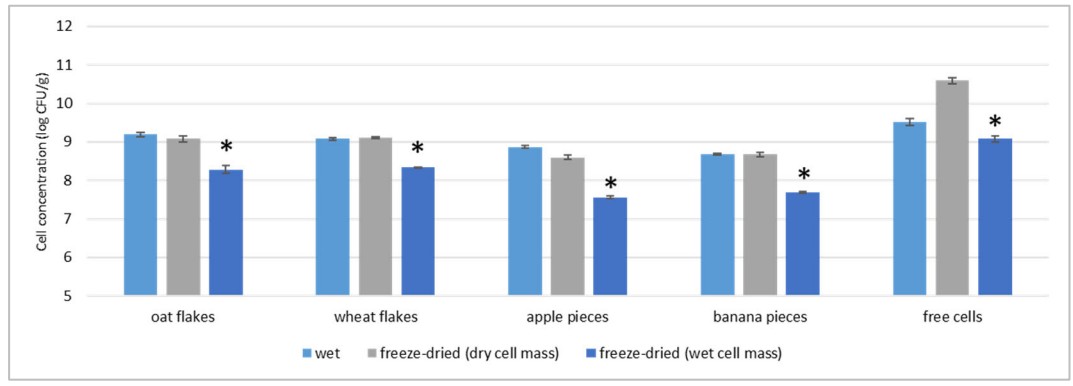

**Figure 2.** Levels (log CFU/g) of free or immobilized *L. rhamnosus* OLXAL-1 cells on cereals and fruit pieces before and after freeze-drying. "Dry cell mass" refers to the concentration of cells after the freeze-drying process, while "wet cell mass" refers to the concentration of freeze-dried cells in their rehydrated state. * $p < 0.05$ compared to initial wet culture (non-freeze-dried) levels.

Freeze-drying led to a significant ($p < 0.05$) reduction in cell viability when comparing the initial populations of wet free or immobilized *L. rhamnosus* cells to the corresponding rehydrated freeze-dried cultures (wet cell mass); however, in all cases, high levels of immobilized cells (>7 log CFU/g) were determined. Despite this reduction, the freeze-

drying process is considered necessary due to the long-term stability it offers, not only at the cellular level [28], but also for overall product safety, quality and convenience during handling and processing [29]. Similarly, high levels of dried immobilized lactic acid bacteria (LAB) on natural supports have been recorded in previous studies. In particular, Prapa et al. [30] studied the effect of cell immobilization and freeze-drying on three different LAB strains (*Lactiplantibacillus plantarum* B282, *Pediococcus acidilactici* SK and *Lactiplantibacillus plantarum* F4). Freeze-dried immobilized LAB cells on zea (*Zea mays*) flakes, pistachio nuts and raisins were recorded at >7 log CFU/g and remained at >6 log CFU/g when stored at room temperature for 40 days. Additionally, Nikolaou et al. [17] also observed high levels of freeze-dried immobilized *L. rhamnosus* cells on apple pieces, with levels exceeding 9 log CFU/g.

Along with the microbiological analysis, cell immobilization was verified by scanning electron microscopy (Figure 3). Interestingly, *L. rhamnosus* OLXAL-1 cells exhibited adhesion to the food matrices and also formed aggregates, previously described as "cell matting" [31].

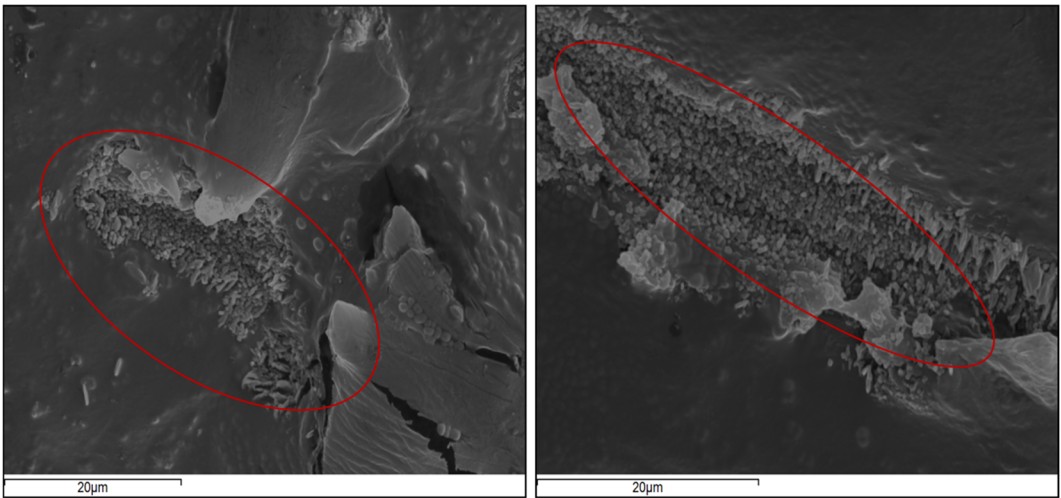

**Figure 3.** Electron micrographs showing (in red circles) immobilized *L. rhamnosus* OLXAL-1 cells on apple pieces (**left**) and oat flakes (**right**).

### 3.2. Cell Survival during In Vitro Digestion Model

To investigate the potential protective effect of cell immobilization on the viability of *L. rhamnosus* OLXAL-1 cells during GI transit, both wet and freeze-dried immobilized cultures were subjected to a static in vitro digestion model. For comparison purposes, wet and freeze-dried free *L. rhamnosus* OLXAL-1 cells were also tested in the same digestion process.

According to Table 1, incubation in the simulated salivary phase had no effect on the viable cell counts of wet and freeze-dried free or immobilized *L. rhamnosus* OLXAL-1 cultures [7,32], while incubation in the simulated gastric and intestinal phase led to a significant ($p < 0.05$) reduction in all cases. Notably, the survival rates (%) of all immobilized *L. rhamnosus* OLXAL-1 cultures after in vitro simulated digestion were higher ($p < 0.05$) compared to free cells. Additionally, higher survival rates were observed in the wet compared to the corresponding freeze-dried cultures for both free and immobilized cells, a result that can be explained by the cellular stress during freeze-drying [33]. Furthermore, statistical analysis showed strong interactions between the state (wet or freeze-dried) and the nature (immobilized or free) of *L. rhamnosus* OLXAL-1 cells regarding their survival during in vitro digestion.

**Table 1.** Tolerance of immobilized and free *L. rhamnosus* OLXAL-1 cultures to simulated digestion phases.

| *L. rhamnosus* OLXAL-1 Cell Cultures | Survival Rate (%) | | |
|---|---|---|---|
| | Simulated Salivary Phase | Simulated Gastric Phase | Simulated Intestinal Phase |
| Wet immobilized cells on oat flakes | 99.39 ± 0.19 [A] | 88.63 ± 0.31 [Be] | 84.76 ± 0.56 [Cf] |
| Freeze-dried immobilized cells on oat flakes | 99.22 ± 0.46 [A] | 82.78 ± 0.26 [Ba] | 81.22 ± 1.36 [Bef] |
| Wet immobilized cells on wheat flakes | 98.42 ± 0.48 [A] | 83.05 ± 0.19 [Bab] | 77.66 ± 1.94 [Bde] |
| Freeze-dried immobilized cells on wheat flakes | 99.46 ± 0.76 [A] | 82.43 ± 0.37 [Ba] | 76.08 ± 0.46 [Cd] |
| Wet immobilized cells on apple pieces | 99.09 ± 1.59 [A] | 87.38 ± 1.10 [Bbe] | 51.01 ± 0.40 [Cb] |
| Freeze-dried immobilized cells on apple pieces | 98.87 ± 1.20 [A] | 83.54 ± 0.09 [Bab] | 48.65 ± 1.35 [Cb] |
| Wet immobilized cells on banana pieces | 98.74 ± 0.16 [A] | 72.13 ± 2.36 [Bd] | 70.01 ± 0.32 [Bc] |
| Freeze-dried immobilized cells on banana pieces | 98.54 ± 0.57 [A] | 71.49 ± 1.22 [Bd] | 66.80 ± 0.18 [Cc] |
| Wet free cells | 99.33 ± 0.22 [A] | 65.74 ± 0.10 [Bc] | 36.29 ± 1.35 [Ca] |
| Freeze-dried free cells | 99.23 ± 0.77 [A] | 65.10 ± 0.85 [Bc] | 33.42 ± 0.86 [Ca] |

Data are presented as mean ± SD. Significant differences ($p < 0.05$) are denoted by different uppercase letters in superscript (row) and different lowercase letters in superscript (column).

In detail, after in vitro digestion, the survival rates of wet and freeze-dried free cells were determined at 36.29% and 33.42%, respectively. Among the wet immobilized cultures, the survival rates ranged from 51.01% to 84.76% and the highest ($p < 0.05$) survival rate was noted in wet immobilized cells on oat flakes, while the lowest ($p < 0.05$) rate was observed in wet immobilized cells on apple pieces. A similar pattern was also observed in the freeze-dried immobilized cultures, with the highest ($p < 0.05$) survival rate recorded in freeze-dried immobilized cells on oat flakes (81.22%) and the lowest ($p < 0.05$) rate in freeze-dried immobilized cells on apple pieces (48.65%).

Resistance to the harsh and degradative conditions of the human GI tract is considered an essential prerequisite for probiotic cultures, in order to confer potential health benefits [34,35]. This requirement is particularly important for LAB strains as they are generally characterized by a poor ability to survive such challenging conditions [36,37]. Several strategies have been proposed to enhance the survival and viability of bacterial cells in the GI tract [38]. Among these strategies, a notable approach is cell immobilization, which involves physically or biochemically confining probiotic cells, preventing their unrestricted movement and providing them with a protective matrix or support structure [39]. In particular, cell immobilization on food matrices/natural carriers has been reported to offer distinct advantages, mainly due to their food-grade nature, which ensures safety compliance, their low-cost and high availability, and, finally, their ease of use for processing during the immobilization process [13].

Several studies have reported a protective effect when comparing immobilized cells on food matrices to free cells after in vitro exposure to low pH, bile salts and digestive enzymes. For instance, in the study conducted by Kyereh et al. [34], *L. plantarum* NCIMB 8826 cells were immobilized on a weanimix (cereal–legume blend) food matrix and then exposed to two separate tolerance assays: one against low pH and another against the presence of bile salts. In both cases, the weanimix food matrix provided a protective effect on *L. plantarum* cells, as significantly higher survival rates were determined in immobilized cultures compared to free cells after in vitro digestion. In the same manner, Michida et al. [40] reported that cell immobilization on barley and malt fibers contributed to the stability of LAB cells after exposure to simulated gastric and bile juices. Apart from

the use of cereals as immobilization supports, various fruits have also been studied for their potential protective effect against simulated GI transit. Vitola et al. [41] studied the immobilization of *Lactobacillus casei* CSL3 on kiwi, guava and pineapple pieces, which were subsequently incorporated into *Petit Suisse* cheese. Notably, *Petit Suisse* cheeses with either immobilized or free cells exposed to simulated GI transit conditions did not differ in terms of cell viability, a result that, according to the authors, can be attributed to the ingredients included in the food matrix (cheese) that contributed to the preservation of the probiotic bacteria.

According to the abovementioned studies, the observed protective effect of cell immobilization on bacterial cells can be attributed to various factors. One of the primary contributors, which has been highlighted in several studies, is the buffering capacity of ingested food, which transiently increases the pH values in gastric juices, thereby providing a protective shield for probiotic bacteria [36]. Furthermore, the presence of fibers and sugars in cereals and fruits has also been suggested as contributing agents, since they offer an optimal substrate for the growth and survival of LAB [42–44]. Additionally, it is also suggested that the attachment of cells to the food matrix, or their entrapment within it, acts as a barrier, reducing the susceptibility of probiotics to adverse conditions during GI transit [45]. Nevertheless, to validate these observed protective effects gained from in vitro studies, it is essential to further investigate the effectiveness of cell immobilization on food matrices in a dynamic in vivo model.

### 3.3. Cell Survival and Attachment of L. rhamnosus OLXAL-1 Cells through an In Vivo Mouse Model

Taking into account the results of the in vitro digestion model, the freeze-dried immobilized cell culture exhibiting the highest survival rate was chosen for further assessment within an in vivo BALB/c mouse model. Freeze-dried immobilized *L. rhamnosus* OLXAL-1 cells on oat flakes demonstrated a survival rate of 81.22%, trailing slightly behind the wet immobilized *L. rhamnosus* OLXAL-1 cells on oat flakes, with a survival rate of 84.76%. The selection of the freeze-dried culture, despite the lower survival rate, was based on industrial and commercial requirements for stability and convenience. For comparison purposes, three control mice groups were also included in the study (CD, OD and FD), in order to assess the effect of oat flakes on lactobacilli counts in feces, intestinal fluid and intestinal tissue.

According to the results (Figure 4), lactobacilli levels in mice feces were encountered at levels ranging from 7.00 to 7.12 log CFU/g [46] at baseline level (day 0) in all groups. Following administration of both immobilized and free *L. rhamnosus* OLXAL-1 cells (groups ID and FD), a significant ($p < 0.05$) increase in lactobacilli populations was observed in mice feces after the 10-day period compared to baseline. A significant ($p < 0.05$) increase in lactobacilli counts was also observed in the feces of the ID group compared to the FD group after the 10-day *L. rhamnosus* administration (8.02 log CFU/g and 7.64 log CFU/g, respectively). In contrast, in the control groups CD and OD, which received a control diet and an oat-flakes-supplemented control diet, respectively, stable lactobacilli counts at approximately 7 log CFU/g were determined, indicating that neither the rodent chow nor the oat flakes supplementation affected lactobacilli numbers.

A similar pattern was also observed in the intestinal fluids of mice obtained from the ileum, cecum and colon after euthanasia. According to Table 2, lactobacilli counts in the intestinal fluid samples derived from the ileum, cecum and colon were higher ($p < 0.05$) in groups ID and FD compared to the control groups (CD and OD). Additionally, an increase in lactobacilli counts ($p < 0.05$) was observed in group ID compared to group FD, aligning with the findings of Figure 4. Higher ($p < 0.05$) lactobacilli counts were also determined in the tissue samples (ileum, cecum and colon) in groups ID and FD compared to control groups CD and OD. However, it is noteworthy that, although higher lactobacilli counts were observed in both the intestinal fluids and feces of the ID group compared to the

FD group, this difference was not observed in the intestinal tissue specimens, where the lactobacilli counts did not differ ($p > 0.05$) between these two groups.

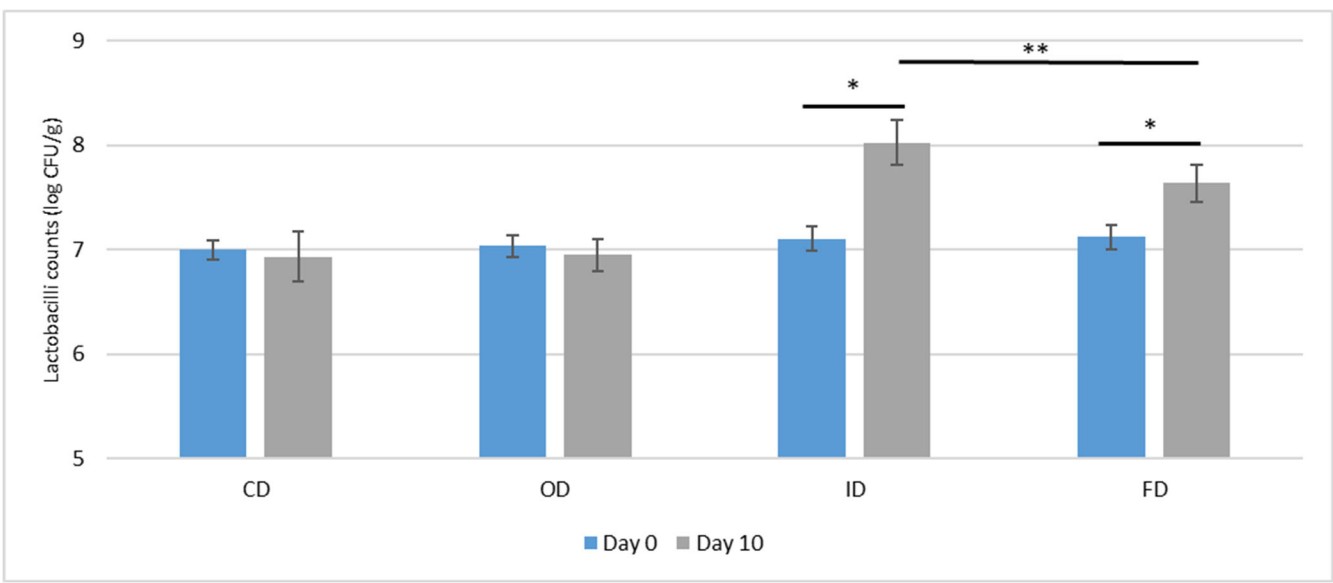

**Figure 4.** Levels (log CFU/g) of lactobacilli in BALB/c mouse feces before (day 0) and after (day 10) the administration of freeze-dried free or immobilized *L. rhamnosus* OLXAL-1 cells on oat flakes. Data are expressed as mean $\pm$ SD (n = 6 per group). * $p < 0.05$ compared to day 0; ** $p < 0.05$ when comparing ID vs. FD groups (day 10). CD: animals that received the control diet; OD: animals that received the control diet supplemented with a daily dose of 2 g of freeze-dried oat flakes; ID: animals that received the control diet supplemented with a daily dose of 2 g of freeze-dried immobilized *L. rhamnosus* OLXAL-1 cells on oat flakes; FD: animals that received the control diet supplemented with freeze-dried free *L. rhamnosus* OLXAL-1 cells.

**Table 2.** Populations (log CFU/g) of lactobacilli in BALB/c mouse intestinal tissue and fluid samples after the administration of freeze-dried free or immobilized *L. rhamnosus* OLXAL-1 cells on oat flakes.

|  | CD | OD | ID | FD |
|---|---|---|---|---|
| Tissue | Lactobacilli counts (log CFU/g) | | | |
| Ileum | 5.02 $\pm$ 0.19 [a] | 5.04 $\pm$ 0.15 [a] | 5.82 $\pm$ 0.18 [b] | 5.84 $\pm$ 0.11 [b] |
| Cecum | 5.09 $\pm$ 0.16 [a] | 5.04 $\pm$ 0.11 [a] | 6.25 $\pm$ 0.20 [b] | 6.13 $\pm$ 0.37 [b] |
| Colon | 4.95 $\pm$ 0.20 [a] | 5.02 $\pm$ 0.14 [a] | 6.21 $\pm$ 0.15 [b] | 6.11 $\pm$ 0.31 [b] |
| Intestinal fluid | | | | |
| Ileum | 4.97 $\pm$ 0.36 [a] | 5.18 $\pm$ 0.41 [a] | 6.96 $\pm$ 0.20 [b] | 6.42 $\pm$ 0.29 [b] |
| Cecum | 6.93 $\pm$ 0.22 [a] | 6.95 $\pm$ 0.18 [a] | 7.97 $\pm$ 0.18 [b] | 7.64 $\pm$ 0.16 [c] |
| Colon | 6.94 $\pm$ 0.14 [a] | 6.88 $\pm$ 0.14 [a] | 8.01 $\pm$ 0.16 [b] | 7.54 $\pm$ 0.24 [c] |

Data are expressed as mean $\pm$ SD (n = 6 per group). Significant differences ($p < 0.05$) in the same row are indicated by different letters in superscript. CD: animals that followed the control diet; OD: animals that followed the control diet supplemented with a daily dose of 2 g of freeze-dried oat flakes; ID: animals that received the control diet supplemented with a daily dose of 2 g of freeze-dried immobilized *L. rhamnosus* OLXAL-1 cells on oat flakes; FD: animals that received the control diet supplemented with freeze-dried free *L. rhamnosus* OLXAL-1 cells.

Reviewing the results presented in Figure 4 and Table 2, it can be clearly suggested that the oral administration of *L. rhamnosus* OLXAL-1 cells, either in immobilized or free form, led to a significant rise in lactobacilli counts in the intestinal epithelium, fluids and mice feces. In contrast, the inclusion of oat flakes (with no *L. rhamnosus* OLXAL-1 cells) in the mice's diet did not yield the same outcome. Furthermore, the implementation of cell immobilization seemed to positively influence the survival, but not the attachment,

of *L. rhamnosus* cells when exposed to in vivo GI transit due to the increased lactobacilli counts in group ID compared to group FD.

As a next step, and in order to confirm that the observed rise of lactobacilli counts in groups ID and FD occurred as a result of *L. rhamnosus* OLXAL-1 cells administration, a molecular analysis based on multiplex PCR methodology was applied (Figure 5).

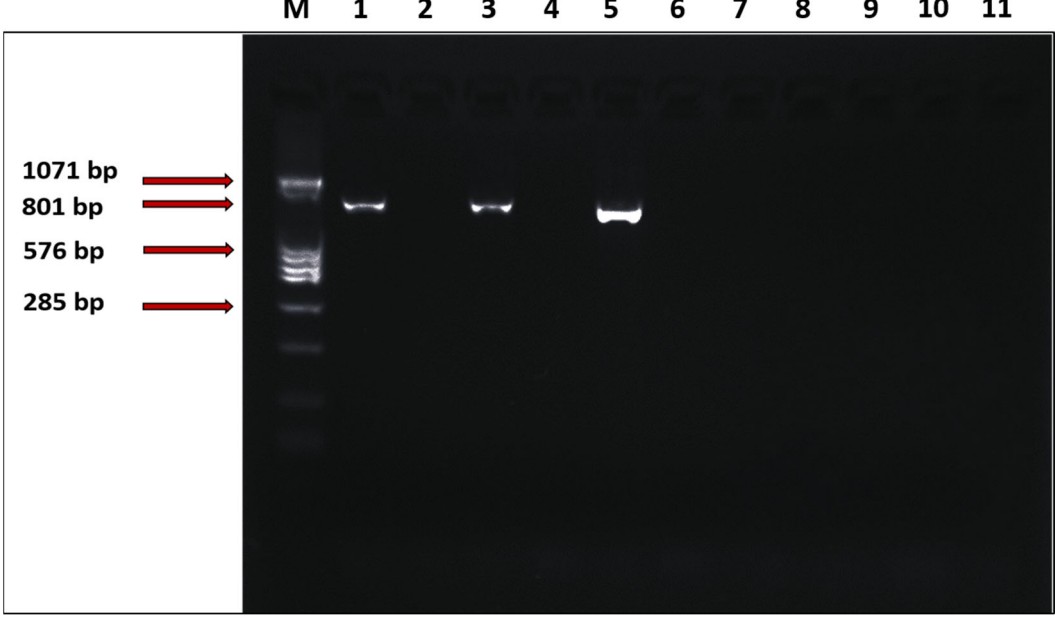

**Figure 5.** Presence of *L. rhamnosus* at levels > 6 log CFU/g by multiplex PCR in fecal samples before and after the implementation of the 10-day dietary intervention protocol. Lane M: 1 kb DNA ladder (Takara, Shiga, Japan); Lane 1: PCR amplified product from a pure culture of *L. rhamnosus* GG (positive control); Lane 2: PCR amplified product derived from genomic DNA extraction of fecal samples from the FD group on day 0; Lane 3: PCR amplified product derived from genomic DNA extraction of fecal samples from the FD group on day 10; Lane 4: PCR amplified product derived from genomic DNA extraction of fecal samples from the ID group on day 0; Lane 5: PCR amplified product derived from genomic DNA extraction of fecal samples from the ID group on day 10; Lane 6: PCR amplified product derived from genomic DNA extraction of fecal samples from the CD group on day 0; Lane 7: PCR amplified product derived from genomic DNA extraction of fecal samples from the CD group on day 10; Lane 8: PCR amplified product derived from genomic DNA extraction of fecal samples from the OD group on day 0; Lane 9: PCR amplified product derived from genomic DNA extraction of fecal samples from the OD group on day 10; Lane 10: negative control.

The presence of *L. rhamnosus* cells was confirmed based on species-specific primers for the *Lacticasebacillus rhamnosus* group by targeting the *mutL* locus and utilizing *Lacticasebacillus rhamnosus* GG as the reference strain. PCR products of 801 bp confirmed the presence of *L. rhamnosus* cells at levels > 6 log CFU/g in mice feces only in groups ID and FD at day 10 of the experimental protocol, further indicating the survival of the administered cells.

Previous studies which evaluated the viability of LAB cells after GI transit using an in vivo BALB/c mouse model have reported similar results regarding the increase in lactobacilli counts in mice feces. Dehghani et al. [46] investigated the viability of two *Lacticaseibacillus rhamnosus* (M1 and RHM) and four *Lactiplantibacillus plantarum* (M8, a7, LF57 and PIF) strains and reported that after 7 days of daily oral administration ($1 \times 10^9$ CFU per day) the number of LAB increased significantly compared to baseline, reaching levels > 7 log CFU/g of mice feces in all cases. Accordingly, Jeong et al. [47] studied the daily oral administration of *Lactobacillus kefiranofaciens* DN1 ($2 \times 10^8$ CFU per day) to 4-week-old pathogen-free female BALB/c mice for a consumption period of 2 weeks and reported a

rise in lactobacilli populations from 6.62 log CFU/g (baseline) to 7.49 log CFU/g of mice feces at the end of the intervention.

Regarding the evaluation of the potential protective effect of cell immobilization on LAB cells during in vivo GI transit, there have been limited studies that have considered both the inclusion of free and immobilized cultures. In particular, Sidira et al. [48] and Saxami et al. [24] evaluated the tolerance of *Lactobacillus casei* ATCC 393 cells after GI transit and their ability to adhere to the intestinal mucosa, respectively, using a Wistar rat model. Both studies performed immobilization of *L. casei* cells on apple pieces and subsequently incorporated them in a fermented milk product with a final cell concentration of approximately 9 log CFU/g. Fermented milk samples with free or immobilized *L. casei* cells showed no differences regarding the lactobacilli counts in rat feces, and both free or immobilized *L. casei* cells were detected by multiplex PCR analysis at levels > 6 log CFU/g [48]. This result was also verified by Saxami et al. [24], in which the authors additionally confirmed the presence of *L. casei* ATCC 393 cells in all the intestinal specimens tested, without, however, observing significant differences between free and immobilized cells.

*3.4. Comparison of In Vitro vs. In Vivo Digestion Models*

Based on our findings, the immobilized cell cultures exhibited high viability levels (>6 log CFU/g) following both in vitro and in vivo digestion. However, it is important to emphasize that, while a prominent protective effect of cell immobilization was observed in the simulated in vitro digestion model compared to free cells, this level of effectiveness was not present in the in vivo model. Specifically, after in vitro digestion, freeze-dried free *L. rhamnosus* OLXAL-1 cells demonstrated a survival rate of 33.42%, resulting in a final cell concentration of 3.04 log CFU/mL, which was substantially lower than the viability of the corresponding free cultures following in vivo digestion (>6 log CFU/g of mice feces). Despite this discrepancy, cell immobilization exerted a strong influence on the rise of lactobacilli levels in the intestinal fluids and feces compared to free cells, a result indicating an improvement in cell viability.

In general, in vitro methods have long been favored in the field of probiotics research due to their simplicity, cost-effectiveness and capacity to evaluate multiple microbial strains concurrently. However, the ability of presumptive probiotic strains to survive under the challenging conditions of the GI tract has, for the most part, been studied by outdated in vitro tests that inadequately replicate the successive stress conditions that occur in the human body [15]. For instance, conventional in vitro static models employ constant pH levels in simulated gastric fluids, whereas, in reality, the pH levels in the stomach can fluctuate between 1–2 and rise to 4–5 following food consumption. In the same manner, in vitro tests assessing bacterial cell tolerance to bile salts often fall short in simulating the dynamic levels of bile concentration found in an in vivo model. Moreover, most tests commonly utilize porcine or bovine bile, which may not have the same impact on cell viability as human bile [49].

In our study, we sought to address the limitations of traditional in vitro digestion models by employing a more complex approach based on the highly cited study by Minekus et al. [20]. By doing so, we aimed to gain a more comprehensive understanding of the variations observed in the survival rates of free and immobilized cell cultures among the two digestion models applied. Finally, even though discrepancies between the two models were observed, these results align with the existing literature that describes the challenges in assessing the survival of probiotic bacteria in vitro and the importance of in vivo validation.

## 4. Conclusions

The present study suggested the effectiveness of cell immobilization on cereals and fruits as delivery vehicles for *L. rhamnosus* OLXAL-1 cells during GI transit. Notably, after in vitro digestion, all immobilized *L. rhamnosus* OLXAL-1 cultures exhibited higher survival rates compared to free cells, with the highest survival rate recorded for the immobilized

cells on oat flakes (84.76%). Upon administration of both freeze-dried free and immobilized *L. rhamnosus* OLXAL-1 cells to BALB/c mice (groups ID and FD), a significant increase in lactobacilli populations was observed in the mice feces compared to baseline. Importantly, the ID group displayed a significantly higher lactobacilli count than the FD group (8.02 log CFU/g and 7.64 log CFU/g, respectively). Finally, the presence of *L. rhamnosus* cells at levels exceeding 6 log CFU/g was confirmed in the mice feces (groups ID and FD) using multiplex PCR analysis. These findings highlighted the successful survival of immobilized *L. rhamnosus* OLXAL-1 cells and suggested the potential for applications of this approach in the probiotic industry.

**Author Contributions:** Conceptualization, A.E.Y., N.K. and Y.K.; methodology, N.K., G.N., Y.K. and A.E.Y.; formal analysis, G.N., I.P., A.N., G.M., A.E.Y. and Y.K.; investigation, G.N., I.P., A.N. and G.M.; resources, N.K., Y.K. and A.E.Y.; writing—original draft preparation, G.N. and Y.K.; writing—review and editing, I.P., A.N., G.M., N.K., A.E.Y. and Y.K.; supervision, Y.K. and A.E.Y.; funding acquisition, Y.K. All authors have read and agreed to the published version of the manuscript.

**Funding:** The research work was supported by the Hellenic Foundation for Research and Innovation (H.F.R.I.) under the "First Call for H.F.R.I. Research Projects to support Faculty members and Researchers and the procurement of high-cost research equipment grant" (Project Number: HFRI-FM17-2496, acronym: iFUNcultures).

**Institutional Review Board Statement:** Animal experimentation was reviewed and approved by the Veterinary Directorate of the Athens Prefecture (Ref. Number 483166/30-05-2022) and conducted in compliance with the European Directive 2010/63.

**Informed Consent Statement:** Not applicable.

**Data Availability Statement:** The data presented in this study are available on request from the corresponding author. The data are not publicly available due to restrictions of the funding authorities.

**Acknowledgments:** We would like to thank George Skavdis (Laboratory of Molecular Regulation and Diagnostic Technology Development, Department of Molecular Biology and Genetics of the Democritus University of Thrace) and Athanasia Gkika for technical support on species-specific multiplex PCR analysis. We also wish to thank Katerina Govatsi (Laboratory of Electron Microscopy and Microanalysis, School of Natural Sciences, University of Patras) for the SEM images.

**Conflicts of Interest:** The authors declare no conflict of interest.

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
