# Peer review of "Cereals and Fruits as Effective Delivery Vehicles of Lacticaseibacillus rhamnosus through Gastrointestinal Transit"

_applsci, doi:10.3390/app13158643_

Round 1

Reviewer 1 Report

Dear authors,

The manuscript entitled “Cereals and fruits as effective delivery vehicles of Lacticaseibacillus rhamnosus through gastrointestinal transit” is an interesting topic that could be of interest for readers. However, I consider that the article could be accepted after some corrections.

Comments in general

Throughout the document there are many sentences that should be written impersonally.

Abstract

Line 17. Please, write in an impersonal form.

Line 17 – 19. Improve redaction. This sentence is not clear

Introduction

Demasiado redundante en las definiciones de probióticos. En su lugar, usted podría describir con más detalle los estudios relacionados con las matrices utilizadas en la cell immobilization

Materials and methods

Line 78. Where did you acquire or buy the strain?

Line 83 and 86. What did you mean by 1/4?

Results and discussion

Line 206. If you mention that there was a range, you should quote that range, not just a value.

Figure 3. Could you show where the immobilized microorganisms are?

Line 249. There must be agreement. In the table it mentions “simulated oral phase”, but in the text it appears as “simulated salivary phase”

Line 272-283. I think that paragraph could fit in the introduction section, since it does not add much to the discussion.

Conclusions

Line 470. Remove “concluding”. That is redundant

Line 470. Please, write in an impersonal form

Author Response

Dear Reviewer,

We sincerely appreciate your valuable feedback and insightful comments on our initial manuscript. Please find below the answers to your comments:

Reviewer 1:

  1. Comments in general: Throughout the document, there are many sentences that should be written impersonally

Answer: The manuscript has been revised and rewritten according to the reviewer’s suggestions.

2 & 3. Line 17. Please, write in an impersonal form. & Line 17 – 19. Improve redaction. This sentence is not clear

Answer: The sentence was rewritten accordingly (lines 17-19 in the revised manuscript).

  1. Line 78. Where did you acquire or buy the strain?

Answer: Information on the origin of the strain has been included in the revised manuscript (lines78-79 & 88).

  1. Line 83 and 86. What did you mean by 1/4?

Answer: It is “¼-strength Ringer’s solution”, the correct term has been added to the revised manuscript.

  1. Line 206. If you mention that there was a range, you should quote that range, not just a value.

Answer: The range values have been added in the revised manuscript, according to the reviewer’s suggestions (lines 218-219).

  1. Figure 3. Could you show where the immobilized microorganisms are?

Answer: According to the reviewer’s suggestion, red circles have been incorporated in Figure 3 in order to show the presence of cells.

  1. Line 249. There must be agreement. In the table it mentions “simulated oral phase”, but in the text it appears as “simulated salivary phase”

Answer: Based on the suggestion of the reviewer, “simulated oral phase” was rewritten as “simulated salivary phase” throughout the revised manuscript (please see Table 1).

  1. Line 272-283. I think that paragraph could fit in the introduction section, since it does not add much to the discussion.

Answer: We understand the reviewer’s suggestion to include the paragraph in the introduction section, as it may not contribute significantly to the discussion. However, we believe that the paragraph serves in connecting the preceding and succeeding paragraphs, aiding the reader's understanding of the flow and coherence of the manuscript. Nevertheless, if the reviewer insists, we are willing to remove the paragraph accordingly.

  1. Line 470. Remove “concluding”. That is redundant.

Answer: The word concluding has been removed.

  1. Line 470. Please, write in an impersonal form

Answer: Based on the suggestion of the reviewer, the text was written in an impersonal form (lines 485-489 in the revised manuscript).

Please find attached the revised manuscript.

Kind regards,

G. Nelios

Reviewer 2 Report

The study described in the paper is interesting. It evaluates the opportunity to use different fruits and cereals as probiotic carries.

The authors may consider the following recommendations:

·         Review the text. There are open signs for quotations, but they are not closed. Some phrases may need to be split and make the appropriate connection between the resulting sentences.

·         Can the authors highlight the motivation for choosing the Lacticaseibacillus rhamnosus OLXAL- 1 cell for their study and not another one? Please include references if possible.

·         The introduction is defined probiotics according to the WHO/FAO as “live microorganisms which when administered in adequate amounts, confer a health benefit on the host”. Can the authors say what the adequate amount is in this case?

  The paper could be considered for publication after minor changes. It has to be revised by the author(s) and resubmitted with the suggested modification specified in the reviewer’s comments.

 Minor editing of English language required.

Author Response

Dear Reviewer,

We sincerely appreciate your valuable feedback and insightful comments on our initial manuscript. Please find below the answers to your comments:

Reviewer 2:

  1. Review the text. There are open signs for quotations, but they are not closed. Some phrases may need to be split and make the appropriate connection between the resulting sentences.

Answer: The text has been modified, according to the reviewer’s suggestions (lines 36, 36, 39-42).

  1. Can the authors highlight the motivation for choosing the Lacticaseibacillus rhamnosus OLXAL-cell for their study and not another one? Please include references if possible.

Answer: Lacticaseibacillus rhamnosus OLXAL-1 strain was studied due to its potential antidiabetic properties. Relevant reference has been added to the revised manuscript (please see lines 78-80).

  1. The introduction is defined probiotics according to the WHO/FAO as “live microorganisms which when administered in adequate amounts, confer a health benefit on the host”. Can the authors say what the adequate amountis in this case?

Answer: In our study, the daily dosage of L. rhamnosus OLXAL-1 administered to BALB/c mice was 2 x 109 cfu. Based on the updated recommendations of International Scientific Association of Probiotics and Prebiotics (ISAPP), a daily amount of 108 cfu can be considered adequate to provide the intended health benefits. Relevant references have been added to the revised manuscript (please see lines 223-225).

All changes have been marked with track changes applied.

Please find attached the revised manuscript.

Kind regards,

G. Nelios
